# UNIFYING SEMI-SUPERVISED AND ROBUST LEARNING BY MIXUP

**Ryuichiro Hataya, Hideki Nakayama**
Graduate School of Information Science and Technology, The University of Tokyo, Tokyo, Japan
{hataya,nakayama}@nlab.ci.i.u-tokyo.ac.jp

## ABSTRACT

Supervised deep learning methods require cleanly labeled large-scale datasets, but collecting such data is difficult and sometimes impossible. There exist two popular frameworks to alleviate this problem: semi-supervised learning and robust learning to label noise. Although these frameworks relax the restriction of supervised learning, they are studied independently. Hence, the training scheme that is suitable when only small cleanly-labeled data are available remains unknown. In this study, we consider learning from bi-quality data as a generalization of these studies, in which a small portion of data is cleanly labeled, and the rest is corrupt. Under this framework, we compare recent algorithms for semi-supervised and robust learning. The results suggest that semi-supervised learning outperforms robust learning with noisy labels. We also propose a training strategy for mixing mixup techniques to learn from such bi-quality data effectively.

## 1 INTRODUCTION

Learning from imperfect data is essential for applying machine learning, especially data-hungry deep learning, to real-world problems. One approach to handling this problem is semi-supervised learning (SSL), where training data consist of a small amount of labeled data and a large amount of unlabeled data. Another approach is robust learning to label noise (RLL), wherein all data are labeled, but some of them are mislabeled.

SSL leverages large unlabeled data to improve the performance of supervised learning on a limited number of labeled data. In the context of deep SSL, one effective method is to train neural networks to maintain consistency for a small perturbation of unlabeled inputs (Miyato et al. (2018); Tarvainen & Valpola (2017); Verma et al. (2018)). Oliver et al. (2018) refers these methods as consistency regularization.

In the RLL setting, learners need to enhance their performance using corrupted labels and avoid the performance deterioration caused by such data. This requirement is particularly important for deep neural networks because they have ample capacity to remember whole samples even if their labels are completely random (Arpit et al. (2017); Zhang et al. (2017a)). To tackle this problem, some methods use a small amount of clean data to estimate noise transition matrix (Xiao et al. (2015); Hendrycks et al. (2018)) or to learn to select possibly correctly-labeled samples (Jiang et al. (2018); Jenni & Favaro (2018)).

Although both SSL and RLL aim to alleviate the limited-data problem, they have been studied independently and evaluated using different benchmarks. However, if only a small amount of clean data is available, they can be regarded as similar problems. In such as situation, *can RLL outperform SSL under the same settings?* This question was our initial motivation to unify these two lines of research.

In this paper, we introduce a generalization of SSL and RLL, based on the concept of trusted data (Charikar et al. (2017); Hendrycks et al. (2018)) in the literature of RLL. More precisely, we assumed that some labels are guaranteed to be clean, and the rest are noisy. The two learning frameworks can be unified by controlling the ratio of corrupted labels to all labels and the noisiness of label corruption.

Using the shared evaluation procedure in Oliver et al. (2018), we compared recent SSL and RLL algorithms using image classification task and found that the existing RLL methods using a small amount of clean data cannot outperform SSL under this setting. This finding suggests that such RLL algorithms cannot use noisy labels effectively.

Therefore, it is necessary to adaptively use SSL and RLL in a data-driven manner. As a baseline learning algorithm, we propose combining the mixup losses for SSL (Verma et al. (2018)) and RLL (Zhang et al. (2017b)); the results obtained are comparable to those of SSL- and RLL-specific methods and indicate the effective use of useful information from noisy labels.

## 2 LEARNING FROM BI-QUALITY DATA

In this section, we describe the setting of learning from bi-quality data, which is a generalization of SSL and RLL. In this formulation, we assume that the given data consist of two parts: trusted data $\mathcal{D}_T$ (for which the labels are always correct) and untrusted data $\mathcal{D}_U$ ( for which some labels might be wrong). Learners are allowed to access the information irrespective of whether each sample is from $\mathcal{D}_T$ or $\mathcal{D}_U$.

Let us denote the ratio of trusted and untrusted data to the entire data $p$ as follows:

$$p = \frac{|\mathcal{D}_T|}{|\mathcal{D}_T| + |\mathcal{D}_U|} \quad (\text{thus } 1 - p = \frac{|\mathcal{D}_U|}{|\mathcal{D}_T| + |\mathcal{D}_U|}). \tag{1}$$

We also introduce quality as

$$q = 1 - \frac{\mathbf{D}(\mathbf{p}_U(y|x)||\mathbf{p}_T(y|x))}{\mathbf{D}(\mathbf{p}(y)||\mathbf{p}_T(y|x))}, \tag{2}$$

where $\mathbf{p}_T(y|x)$ and $\mathbf{p}_U(y|x)$ are the conditional probabilities of labels when inputs of trusted data and untrusted data, respectively, are given, and $\mathbf{D}$ is a divergence between two probability distributions (e.g., Kullback-Leibler divergence). In equation 2, $q \in [0, 1]$. Obviously, $q = 0$ if the labels are completely independent of the input features (i.e., completely random), and $q = 1$ if the labels are clean. We assume that the quality of untrusted data $q_U$ is in $[0, 1]$, and that of trusted data $q_T$ is 1.

We believe that this setting is realistic. Suppose that there are unlabeled data. Under budget constraint, one strategy to label this dataset as training data is to divide the data to two parts and spend most of the budget on high-quality labeling to acquire $D_T$ (e.g., using experts) and the rest on lower-quality labeling to obtain $D_U$ (e.g., using crowdsourcing), where usually $|D_T| \ll |D_U|$.

Under this generalized framework, SSL is equivalent to the particular case where $q = 0$, that is, labels of untrusted data are entirely random and give no useful information. In SSL, such labels are usually ignored, and the data are treated as unlabeled. Meanwhile, RLL without trustable data is another special case, where $p = 0$ and $0 \leq q < 1$. In some studies on RLL, the setting $p > 0$ is used explicitly (Charikar et al. (2017); Li et al. (2017); Hendrycks et al. (2018)) and implicitly (Jiang et al. (2018); Jenni & Favaro (2018)).

Intuitively, when $q$ is relatively high, we can expect some performance gain by utilizing untrusted, but somewhat informative, labels with RLL. On the contrary, when $q$ is almost zero, untrusted data are not informative anymore, and better generalization may be obtained using only standard SSL. A critical problem in practice, however, is that we cannot know the exact quality of untrusted data, and, therefore, we cannot decide the best learning strategy in advance. To handle such data, we need an adaptive mechanism to properly fuse SSL and RLL in a data-driven manner.

To realize this goal, we propose to combine techniques for SSL and RLL. We use the convex combination of loss functions for SSL $\mathcal{L}_{\text{semi}}$ and RLL $\mathcal{L}_{\text{robust}}$. That is, with an additional hyperparameter $\gamma$, the loss function for untrusted data is defined as follows:

$$\mathcal{L}_U = \gamma \mathcal{L}_{\text{robust}} + (1 - \gamma)\mathcal{L}_{\text{semi}}. \tag{3}$$

For $\mathcal{L}_{\text{robust}}$ and $\mathcal{L}_{\text{semi}}$, any loss functions for SSL and RLL can be used [1]. In this study, as a baseline of this setting, we combine *mixup* techniques for SSL (Verma et al. (2018)) and RLL (Zhang et al. (2017b)).

---

[1] Here, each $\mathcal{L}.$ is cross entropy loss.

---

**Algorithm 1** mixmixup: learning from bi-quality data

---

    Prepare trusted set: $\mathcal{D}_T$, untrusted set: $\mathcal{D}_U$
    Initialize neural network $f$
    Set hyper parameters: $\alpha, \beta, \gamma$
    Set loss function: $\mathcal{L}(\cdot, \cdot)$: categorical cross entropy

    **for** $k \in \{0, 1, \ldots, K-1\}$ :
        Sample $\lambda_\alpha \sim \mathrm{Beta}(\alpha, \alpha), \lambda_\beta \sim \mathrm{Beta}(\beta, \beta)$
        Sample $(x_i, y_i), (x_j, y_j)$ from $\mathcal{D}_T$
        $\mathcal{L}_T = \mathcal{L}(f(\lambda_\alpha x_i + (1 - \lambda_\alpha)x_j), \lambda_\alpha y_i + (1 - \lambda_\alpha)y_j)$
        Sample $(x_i', y_i'), (x_j', y_j')$ from $\mathcal{D}_U$
        Predict $y_i'' = f(x_i')$ and $y_j'' = f(x_j')$
        $\mathcal{L}_{\mathrm{robust}} = \mathcal{L}(f(\lambda_\alpha x_i' + (1 - \lambda_\alpha)x_j'), \lambda_\alpha y_i' + (1 - \lambda_\alpha)y_j')$
        $\mathcal{L}_{\mathrm{semi}} = \mathcal{L}(f(\lambda_\beta x_i' + (1 - \lambda_\beta)x_j'), \lambda_\beta y_i'' + (1 - \lambda_\beta)y_j'')$
        $\mathcal{L}_U = \gamma\mathcal{L}_{\mathrm{robust}} + (1 - \gamma)\mathcal{L}_{\mathrm{semi}}$
        Update parameters of $f$ with $\mathcal{L}_T + \sigma(k)\mathcal{L}_U$         ▷ For $\sigma$, see Section 3

---

## 3   MIXMIXUP

mixup is a regularization technique, where neural network models are trained on virtual training pairs $(\tilde{x}, \tilde{y})$, where

$$\tilde{x} = \lambda x_i + (1 - \lambda)x_j, \quad \tilde{y} = \lambda y_i + (1 - \lambda)y_j. \tag{4}$$

Here, $(x_i, y_i)$ and $(x_j, y_j)$ are sampled from training data, and $\lambda \in [0, 1]$ is sampled from Beta distribution. Zhang et al. (2017b) showed that this method alleviates the performance decrease under label corruption.

In Verma et al. (2018), mixup is used for SSL as consistency regularization (Oliver et al. (2018)). Here, for unlabeled inputs $x_i^{\mathbf{U}}, x_j^{\mathbf{U}}$, neural networks are learned to map $\lambda x_i^{\mathbf{U}} + (1 - \lambda)x_j^{\mathbf{U}}$ to $\lambda y_i^{U} + (1 - \lambda)y_j^{U}$, where $y_i^{\mathbf{U}}, y_j^{\mathbf{U}}$ are the predicted outputs corresponding to $x_i^{\mathbf{U}}, x_j^{\mathbf{U}}$.

We show the details in Algorithm 1. The parameters of neural networks are updated with a combination of losses on trusted data $\mathcal{L}_T$ and untrusted data $\mathcal{L}_U$ as $\mathcal{L}_T + \sigma(k)\mathcal{L}_U$. We use sigmoid scheduling $\sigma(k)$ as Verma et al. (2018).

Because this method mixes the mixup losses, hereinafter we refer to our method as *mixmixup*.

## 4   EXPERIMENTS

### 4.1   SETTINGS

We use CIFAR-10 (Krizhevsky (2009)), which has 50,000 of 10-category images. Following the common protocol of SSL in Oliver et al. (2018), we split the training dataset into 45,000 images for training and 5,000 for validation. To simulate bi-quality data, we sample 4,000 examples from the training data as trusted data and the rest as untrusted data. We randomly replace each label of the untrusted samples with another one with a given probability, which is a common protocol in RLL (Zhang et al. (2017b); Jiang et al. (2018)). We use the non-corrupted validation data and test data for hyperparameter tuning and final evaluation, respectively.

Following Oliver et al. (2018) and Verma et al. (2018), we use WRN-28-2 (Wide ResNet 28-2, proposed in Zagoruyko & Komodakis (2016)) as the image classifier. To optimize the network, we use SGD with a learning rate of 0.1, a momentum of 0.9, a weight decay of $1.0 \times 10^{-4}$ and a minibatch size of 256. We train networks for $1.6 \times 10^5$ iterations, following Verma et al. (2018). We implement the model with PyTorch v1.0 (Paszke et al. (2017) ) and tune hyperparameters $\alpha, \beta$ and $\gamma$ in Algorithm 1 with a Bayesian optimization algorithm in Optuna v0.8 [2].

### 4.2   RESULTS

#### 4.2.1   LEARNING FROM BI-QUALITY DATA

According to the settings in Section 4.1, we trained WRN-28-2 with bi-quality data of different quality. We randomly replace 40% and 100% of the labels of the untrusted data to simulate label

---

[2] https://optuna.org/

Table 1: **mixmixup can handle bi-quality data effectively.** Test accuracy of WRN-28-2 trained on bi-quality CIFAR-10. In the columns under TRUSTED and UNTRUSTED, we list the number of samples of trusted and untrusted data with quality $q$.

| | METHOD | TRUSTED | UNTRUSTED | ACCURACY |
|---|---|---|---|---|
| A | Basic | 4,000 | N/A | 0.72 |
| | input mixup (Zhang et al. (2017b)) | 4,000 | N/A | 0.78 |
| B | Basic | 4,000 | 41,000 ($q = 0.0$) | 0.29 |
| | Basic | 4,000 | 41,000 ($q = 0.6$) | 0.78 |
| C | input mixup (Zhang et al. (2017b)) | 4,000 | 41,000 ($q = 0.0$) | 0.43 |
| | input mixup (Zhang et al. (2017b)) | 4,000 | 41,000 ($q = 0.6$) | 0.89 |
| D | mixmixup (ours) | 4,000 | 41,000 ($q = 0.0$) | 0.88 |
| | mixmixup (ours) | 4,000 | 41,000 ($q = 0.6$) | 0.90 |

Table 2: **Semi-supervised methods can surpass robust learning methods under shared settings.** Test accuracy of WRN-28-2 trained on CIFAR-10 with other state-of-the-art methods for semi-supervised learning and robust learning. Here, [†] and [††] refer to the scores in the tables are from Oliver et al. (2018) and our re-implementations, respectively.

| | METHOD | TRUSTED | UNTRUSTED | ACCURACY |
|---|---|---|---|---|
| A | input mixup (Verma et al. (2018)) | 4,000 | 41,000 (no label) | 0.89 |
| | Mean teacher[†] (Tarvainen & Valpola (2017)) | 4,000 | 41,000 (no label) | 0.84 |
| | VAT[†] (Miyato et al. (2018)) | 4,000 | 41,000 (no label) | 0.86 |
| B | MentorNet DD-MLP[††] (Jiang et al. (2018)) | 4,000 | 41,000 ($q = 0.6$) | 0.87 |
| | GLC[††](Hendrycks et al. (2018)) | 4,000 | 41,000 ($q = 0.6$) | 0.84 |
| C | mixmixup (ours) | 4,000 | 41,000 ($q = 0.0$) | 0.88 |
| | mixmixup (ours) | 4,000 | 41,000 ($q = 0.6$) | 0.90 |

corruption. From the definition of quality (equation 2), the quality of these data is 0.6 and 0.0, respectively. The former corresponds to RLL with trusted data, and the latter to SSL.

### 4.2.2 COMPARISON WITH OTHER METHODS

Table 2 presents the test accuracy of SSL methods (Verma et al. (2018); Tarvainen & Valpola (2017); Miyato et al. (2018), Table 2 A) and RLL methods with trusted data (Jiang et al. (2018); Hendrycks et al. (2018), Table 2 B) with the results obtained using mixmixup (Table 2 C). In all the experiments, the same network (WRN-28-2) and the same dataset split mentioned above are used. Note that SSL methods in Table 2 A does not use labels of untrusted data.

Surprisingly, our results suggest that SSL (Table 2 A) can provide better performance than RLL (Table 2 B) with trusted data for identical settings. This result suggests the difficulty of learning robustly from partially corrupted labels preventing overfitting.

Further, mixmixup can handle both SSL and RLL. Moreover, the accuracy when the quality is 0.6 (corresponding to RLL) is superior to that for quality 0.0 (corresponding to SSL), indicating that mixmixup can effectively use information from corrupted labels.

## 5 CONCLUSION

In this paper, we introduce a novel framework of weakly supervised learning by unifying SSL and RLL, which have been independently studied. To handle this problem, we propose to mix *mixup* for SSL and RLL. This method empirically works well and achieves competitive results with semi-supervised and robust learning specific methodologies.

In addition, our experiments indicate that the performance of some RLL with trusted data might be inferior to that of SSL under identical settings. This result suggests that the existing RLL methods cannot effectively exploit the information which should be extracted from noisy labels.

Our proposed method does not use the estimated quality; instead some hyperparameters are introduced. The use of quality estimation may ease hyperparameter tuning, but is still an open question.

### ACKNOWLEDGEMENT

This work was supported by JSPS KAKENHI Grant Number JP19H04166.

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
