# OpenReview forum: "Unifying semi-supervised and robust learning by mixup"
_ICLR.cc/2019/Workshop/LLD — LLD 2019_

### Official Review · AnonReviewer1 · 2019-03-28
**Great workshop paper, but many areas to improve algorithm**

**Rating:** 5
**Confidence:** 2

**Review:**

Major comments and summary:

  Overall, I strongly recommend this paper as a workshop paper, but I think it needs more work to become a great conference paper.  The problem is well-motivated and seems very important to me.  My biggest issue is that I feel like one could go much "further" with this idea - and it shows empirically - the only improvement from using the noisy labels is to go from 89% -> 90%, which I would say is only a small improvement.  If this problem gets solved I think you should be closer to 95% on CIFAR10.

I have a few ideas on how to make the algorithm better:
  -On the noisily labeled data points $D_U$, replace the (noisy) label y_i with a soft-label which reflects the underlying noise process.  For example if you're told that y = 1, but the noise process corrupts to a new random label with 50% chance, then make the new soft-label [0.5/9, 0.5, 0.5/9, 0.5/9, ...].  This is like changing the label randomly each time you see the example but it will have much less variance!  This is just for the L_robust loss.
  -Alternatively, what if you replace L_robust with a partial likelihood which reflects what q is?  For example, if each data point is corrupted with 50% chance - then just try to encourage the probability of that point to be >=50%.  Don't try to push the probability on that label up to 100% (in practice this would look like a hinge loss - I think).
  -Is "L" loss cross-entropy or mean squared error?  For semi-supervised consistency loss I think it helps to use mean squared error instead of cross entropy (cross-entropy is very harsh if the prediction and the label strongly disagree when the label is confident).

Other Comments:

  -I think the bi-quality data setting is extremely interesting.  However, the introduction could perhaps benefit from giving some more concrete examples of bi-quality data.  I think model-based RL could be one interesting example (the model gives very noise rollouts and the environment gives high-quality rollouts).

  -The paper motivates the algorithm by saying that the value of q might be unknown.  However, even if q were known, would it be trivial to fuse SSL and RLL?  There would still be a tradeoff between how much to trust the labels from the untrustworthy set and how much to rely on the trustworthy data.

  -Explicit hyperparameter to mix between the robust objective the semi-supervised objective.  Would it be too hard to learn this hyperparameter?

  -Is the value of "q" and the noise corruption process known?  It doesn't seem to be used in the algorithm block.  If q or the corruption process were known, would you have a way of using it.


Paper reading notes:
-Deep learning requires labeled data which is both large and clean.

-This paper proposes to study the robust learning and SSL learning problems jointly - in that we assume that we have "bi-quality data" - where a small number of examples are cleanly labeled and a large amount of data has noisy labels.

-Using just robust learning on all the data performs worse than SSL.

-Paper proposes to combine SSL and RLL.

-D_T is the trusted data (clean, labels always correct).  D_U is the untrusted data (labels are sometimes wrong).

-P refers to the fraction of the total data which is trusted.

-Also have a score q, such that q=0 means the labels are totally random and q=1 means the labels are perfectly clean.

-From this perspective SSL corresponds to the q=0 case (where the untrusted data has no label information).


Minor comments:

  -Would be good to also cite this newer paper focusing on mixup and SSL (only published recently, but spells the SSL stuff out more clearly): https://arxiv.org/pdf/1903.03825.pdf

  -"To realize this goal, whose quality might be 0, we propose..." -> this line doesn't make sense to me.  Typo?

  -Please put more space into the algorithm 1 block.

---

### Official Review · AnonReviewer2 · 2019-04-07
**A simple approach for handling noisy labels, coupled with promising results**

**Rating:** 4
**Confidence:** 2

**Review:**

The authors start by introducing a formal setting that includes the semi-supervised and the robust learning tasks as special cases and they then proceed by proposing a strategy based on mixup [1] for training a model in this unified setting.

My general impression from this work is firmly positive. The manuscript is clearly written, the experimental setup covers adequately alternative methods and the description of the experiments includes the most important details. The results presented, while in my opinion do not allow drawing definite conclusions, they are at least promising. In more detail, I see the following prons and cons:

Prons:

1. Clearly written manuscript. Related work is properly presented, the proposed unifying framework is easy to understand and the experiments are described in adequate detail.

2. The proposed learning approach in this unified setting is elegant and, to the best of my knowledge, original.

3. The proposed method is compared adequately with alternative state-of-the-art ones and with reasonable baselines. The comparison of these state-of-the-art methods with each other in this specific setting is interesting on its own right, too.

4. This works falls nicely within the scope of the workshop.

Cons:

1. In the reported experiments the proposed method had a performance which is very close to that of an alternative method (90% as oppose to 89% of [2]) while the difference when q=0 with when q=0.6 is also close (88% and 90% respectively). Given how close these numbers are, I would have preferred if the experiments had been repeated for more trials (say 3) and/or some statistical tests had been performed (even though it is my understanding that this is not standard practice in the conference) in order to gain some insight on the statistical significance of these differences.

2. Of course one can always suggest more experiments,  but still I think it would be interesting to see if/how the results change if some other CNN architecture is used.


[1] Zhang, Hongyi, et al. "mixup: Beyond empirical risk minimization." arXiv preprint arXiv:1710.09412 (2017).

[2] Verma, Vikas, et al. "Manifold Mixup: Learning Better Representations by Interpolating Hidden States." (2018).

---

### Decision · Program_Chairs · 2019-04-08
**Acceptance Decision**

Accept